# Free-oscillation technique: The effect of the magnitude of the impulse applied on muscle and tendon stiffness around the ankle

**Aurélio Faria[1], Ronaldo Gabriel[2], Rui Brás[1], Helena Moreira[3], Márcio Soares[1], Massimiliano Ditroilo[4]***

**1** Department of Sport Sciences, CIDESD, University of Beira Interior, Covilhã, Portugal, **2** Department of Sport Sciences, Exercise and Health – CITAB, University of Trás-os-Montes and Alto Douro, Vila Real, Portugal, **3** Department of Sport Sciences, Exercise and Health - CIDESD, University of Trás-os-Montes and Alto Douro, Vila Real, Portugal, **4** School of Public Health, Physiotherapy and Sports Science, University College Dublin, Dublin, Ireland

* massimiliano.ditroilo@ucd.ie

**Data Availability Statement:** Data relevant to this study are available from OSF at https://osf.io/53sjx/files/osfstorage/63d9509a78a62300ec99de98.

## Abstract

The importance of the muscle-tendon complex in sport and for activities of everyday living is well recognised. The free oscillation technique is frequently used to determine the musculo-articular "apparent" stiffness (obtained from vertical ground reaction force) and other parameters. However, an in-depth understanding of the muscle-tendon complex can be gained by separating the muscle (soleus) and the tendon (Achilles tendon) components and studying the "true" stiffness for each of these components (by considering the ankle joint moment arms), which can be valuable in improving our understanding of training, injury prevention, and recovery programs. Hence, this study aimed to investigate if muscle and tendon stiffness (i.e., "true" stiffness) are similarly affected by different impulse magnitudes when using the free-oscillation technique. Three impulse magnitudes (impulse 1, 2 and 3), corresponding to peak forces of 100, 150 and 200 N, were used to estimate the stiffness of the ankle joint in 27 males, using multiple loads (10, 15, 20, 25, 30, 35, and 40 kg). A significant decrease ($p < 0.0005$) was found in musculo-articular "apparent" stiffness (29224 ± 5087 N.m$^{-1}$; 27839 ± 4914 N.m$^{-1}$; 26835 ± 4880 N.m$^{-1}$) between impulses 1, 2 and 3 respectively, when loads were collapsed across groups. However, significant differences ($p < 0.001$) were only found between the median (Mdn) of impulse 1 (Mdn = 564.31 (kN/m)/kN) and 2 (Mdn = 468.88 (kN/m)/kN) and between impulse 1 (Mdn = 564.31 (kN/m)/kN) and 3 (Mdn = 422.19 (kN/m)/kN), for "true" muscle stiffness, but not for "true" tendon stiffness (Mdn = 197.35 kN/m; Mdn = 210.26 kN/m; Mdn = 201.60 kN/m). The results suggest that the musculo-articular "apparent" stiffness around the ankle joint is influenced by the magnitude of the impulse applied. Interestingly, this is driven by muscle stiffness, whereas tendon stiffness appears to be unaffected.

**Funding:** This work was partially supported by the Portuguese Foundation for Science and Technology (https://www.fct.pt/en/), project UID04045/2020. Initials of the authors who received the award: - AF - RB - HM The funder had no role in study design, data collection and analysis, decision to publish, or preparation of the manuscript.

**Competing interests:** The authors have declared that no competing interests exist.

## 1. Introduction

The importance of the muscle-tendon complex in human movement is widely recognized, whether it is associated with sport or activities of everyday living. The elastic behaviour of the muscle-tendon complex is usually studied using spring-mass models and one of the most studied biomechanical properties in this context is stiffness [1–4]. In simple terms, stiffness is defined as the resistance to deformation of an elastic structure when a force is applied to it [5]. Stiffness has been associated with sports performance [6, 7], injury risk [8, 9], as well as stability and postural control [10, 11]. The assessment of musculo-articular stiffness through the free oscillation technique is a widely used method that aims to assess stiffness around a particular joint. Musculo-articular stiffness encompasses, several components, such as muscles, tendons, ligaments, cartilage, fáscia and bone. However, muscle and tendon are the predominant factors [8, 12, 13]. Musculo-articular stiffness of the ankle has received most of the literature attention [2, 4, 13, 14], particularly due to its importance for weight-bearing stability, shock absorption and propulsive force [15]. In the free oscillation technique, the ankle joint supports an isometric load whilst an impulse is applied with the objective of destabilizing the ankle joint and causing the leg to oscillate in the sagittal plane [16]. This oscillation is then measured, and stiffness estimated. Based on the assumption that an elastic system oscillates at its natural or resonant frequency regardless of the magnitude of the impulse applied, most studies applied this impulse manually [5, 13]. However, previous studies suggested that the magnitude of the impulse has the potential to affect co-activation [17] and/or the stretch reflex [18, 19], which in turn can influence stiffness. Based on these considerations, Faria, Gabriel [5] assessed the effect of three impulses of different peak force magnitudes (100, 150 and 200 N) on ankle musculo-articular stiffness. The authors reported that as the impulse magnitude increased, the stiffness of the ankle significantly decreased.

Musculo-articular stiffness is a measure of the whole ankle joint stiffness estimated from the ground reaction forces. When stiffness is calculated directly from ground reaction forces, it is termed 'apparent' stiffness since it relates to the center of mass movement of the system, which is being measured at equilibria [20–22]. However, taking into consideration the moment arms of the ankle joint, it is possible to obtain the 'true' stiffness by separating the stiffness of the soleus ($k_m$) and Achilles tendon ($k_t$) [2, 20]. Although knowledge of 'apparent' stiffness might be adequate to propose training programs and estimate training levels, an in-depth understanding of muscle and tendon individually will allow a more thorough control of the training process, injury prevention protocols and recovery assessments [2, 23]. It is recognized that to optimize an athlete's performance and/or rehabilitation, training programs should be developed considering the long-term goals of the athlete, the state of recovery and performance, the type and frequency of injuries, as well as the functional and architectural requirements for the muscle and tendon [24]. Muscles and tendons are both frequently injured simultaneous due their anatomical relationship, however specific injuries can also occur separately. Detailed information of muscle and tendon in training or injury recovery stage is therefore important for decision making. Training programs also need to account for the specificity of tendons since tendon responses to training have been reported to be slower than those in muscles [25]. Furthermore, $k_t$ on its own may provide relevant information to track the evolution of the effects of training and the recovery intervention [23]. In estimating $k_m$ and $k_t$ using the free oscillation technique, it is therefore important to understand whether the technique itself affects the determination of stiffness of these two components. Therefore, this study aimed to compare muscle and tendon stiffness outcomes between the three impulses applied and determine how muscle and tendon are affected by impulse magnitudes. It was hypothesized that different impulse magnitudes would affect the stiffness of the muscle and tendon differently.

## 2. Methods

### 2.1 - Participants

Twenty-seven male university students (age 20.7 ± 1.3 years; height 1.73 ± 0.05 m; body mass 74.7 ± 8.8 kg) healthy and with no history of ankle injuries volunteered for the study. Participants were informed of the scope of the study, and written informed consent was obtained. The study was approved by the ethics committee of the University of Beira Interior (Portugal) and conducted in accordance with the Declaration of Helsinki.

### 2.2 - Impulse calibration

Impulse calibration was performed as previously detailed in Faria, Gabriel [5]. The following results were obtained: Peak force for impulse 1 = 100 ± 0.85 N, peak force for impulse 2 = 150 ± 0.95 N and peak force for impulse 3 = 200 ± 0.86 N.

### 2.3 - Stiffness assessment

Musculo-articular stiffness was evaluated using the free oscillation technique, where participants sat in the position illustrated in Fig 1a, with their arms crossed over their chests and without their thighs supported. The position was adjusted so that the foot-leg, leg-thigh and thigh-trunk angles were 90 deg while maintaining the metatarsophalangeal joint of the right foot (barefoot) aligned with the edge of the metal block (C) placed on top of the force plate (Kistler 9281B; Kistler Instruments, Amherst, NY, USA). The calcaneus was also supported on the metal block (D) to minimize the effects of fatigue until the setup was fully prepared for stiffness evaluation. Only the stiffness of the right lower limb was evaluated because published studies suggest that there are no significant stiffness differences between lower limbs [26]. The maintenance of the foot position between measurements was ensured through mechanism (A), which allows measuring the distance between the end of the toes and the end of the metal

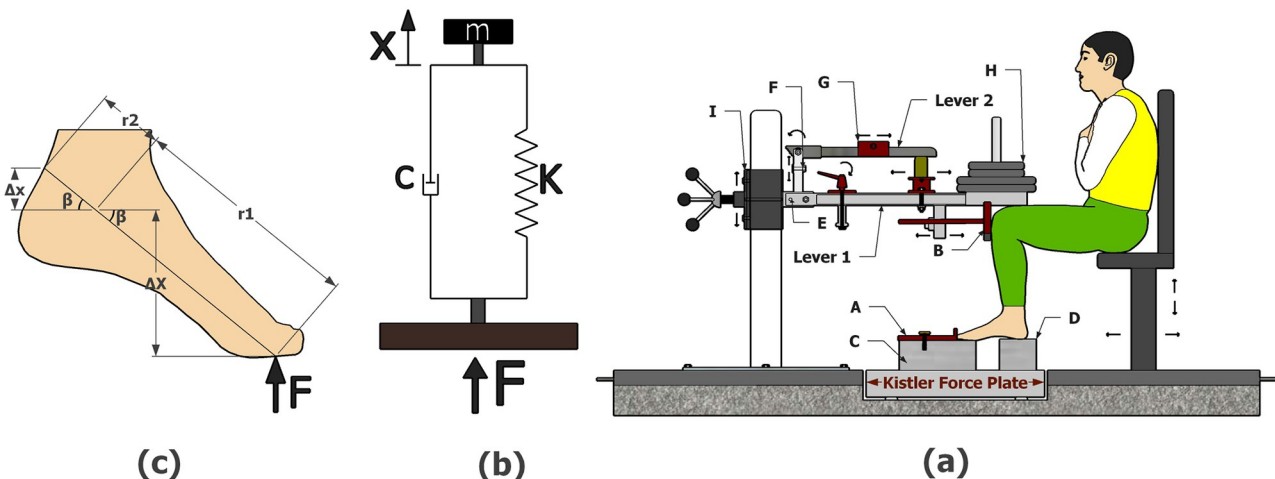

**Fig 1.** Panel (a) shows the equipment and position used to assess the stiffness of the ankle joint: (A) to lock the foot in position, (B) to lock the knee in position, (C and D) metal blocks used to support the foot, (E) to lock lever 1, (F) to set the initial position of lever 2, (G) weight to calibrate lever 2 that can be moved to left and right, (H) standard weights (loads), (I) to adjust lever 1 height. Panel (b) shows the mass-spring model of the system: (F) reaction force, (m) mass of the system, (K) "apparent" stiffness, (C) "apparent" damped coefficient and (X) displacement of the mass. Panel (c) the diagram of the moment arms: The reaction force is represented by (F), the displacement around the ankle by (Δx and ΔX), and the distance between ankle joint-forefoot and ankle joint-rearfoot by (r1) and (r2) respectively.

block (C). Then, lever 1 was supported at 10 cm away from the knee, and this position was controlled through mechanism (B) for all stiffness evaluations. A spirit level was used to keep lever 1 horizontal. To assess the stiffness, seven external loads (i.e., 10, 15, 20, 25, 30, 35, and 40 kg) were randomly placed individually on top of lever 1. For each load, the stiffness was evaluated using each of the three calibrated impulses. Before starting the measurements, lever 1 was unlocked through mechanism (E), and the metal block (D) was removed to allow the calcaneus to oscillate in the sagittal plane during stiffness measurement. Participants were instructed to perform an isometric contraction keeping the ankle at 90 deg and sustain lever 1 with the load horizontally for ~10s. Each of the three calibrated impulses (1, 2, and 3) was randomly applied by dropping lever 2 onto lever 1 and generating a peak force of 100, 150, and 200N, respectively. After each impact, lever 2 bounced, and the investigator grabbed it before it touched lever 1 again. Additionally, being blindfolded, participants were instructed not to react to any stimulus during assessments [12, 18]. Ground reaction forces were collected at the same time with the software Bioware (type 2812A, Version 4.0, Amherst, NY) and Acqknowledge (Version 3.9.1, Montreal, QC, Canada), which was used in conjunction with the Biopac hardware (MP100, Montreal, QC, Canada). The ankle oscillations produced by the impulse were recorded using the force plate and used to assess stiffness. Five trials were performed for each impulse, while rest periods of 2–5 minutes were taken between measurements to avoid fatigue. The average of the five trials was used to calculate the "apparent" stiffness. Additionally, for the "true" stiffness five trials were used to estimate by the nonlinear least squares method $k_m$ and $k_t$, as explained in the following sections.

## 2.4 - Data analysis

Based on the model illustrated in (Fig 1b) the stiffness K and the damping coefficient C around the ankle joint were evaluated. These variables calculated from the ground reaction forces are generally referred to as "apparent" stiffness, as they are related to the vertical displacement of the centre of mass of the body considered into the analysis. However, if we consider the moment arms $r_1$ and $r_2$ (Fig 1c), we can obtain the "true" stiffness and the damping coefficient that will be described from now on in lowercase and in italics as (*k*) and (*c*), respectively [2, 20].

**2.4.1 - "Apparent" stiffness.** The equation of motion used to model the damped mass-spring system, illustrated in (Fig 1b), was Eq 1,

$$(F =)ma = -\text{Kx} - \text{Cv} = 0 \tag{1}$$

which reorganized into Eq 2.

$$m\ddot{\text{x}} + \text{Kx} + \text{C}\dot{\text{x}} = 0 \tag{2}$$

where C represents the "apparent" damping coefficient (Ns/m), $\dot{\text{X}}$ the velocity (m/s), K the "apparent" stiffness (N/m), X the displacement (m), *m* the total mass (kg) of the system (foot + leg + thigh + lever + standard weights) and $\ddot{\text{X}}$ acceleration (m/s$^2$). For an underdamped system, the solution to this equation is Eq 3:

$$X(t) = e^{-\gamma t}(A_x \cos \omega_d t + B_x \sin \omega_d t) \tag{3}$$

where $A_x$ and $B_x$ are integration constants.

Eqs 4 to 6 were used to evaluate the parameter $\gamma$, the damped angular frequency of oscillation $\omega_d$ and the undamped angular frequency of oscillation $\omega_0$.

$$\gamma = \frac{C}{2m} \tag{4}$$

$$\omega_d = \sqrt{\omega_0^2 - \gamma^2} \tag{5}$$

$$\omega_0 = \sqrt{\frac{K}{m}} \tag{6}$$

Assuming the mass of the system is represented by the block $m$ in (Fig 1b), the relationship between X and the ground reaction force F is:

$$m\ddot{x} = F - mg \tag{7}$$

where the acceleration due to gravity is represented by $g$. By replacing Eq 3 into Eq 7 we obtain Eq 8.

$$F(t) = e^{-\gamma t}(A_F \cos \omega_d t + B_F \sin \omega_d t) + mg \tag{8}$$

where $A_F$ and $B_F$ are integration constants.

It is possible to estimate the mass of the system before the application of the impulse (measuring the mass of the foot + leg + thigh + lever + standard weights) or after the impulse (measuring the steady state signal of the force-time curve before or after the oscillation). However, some micro-movements and oscillations may occur, thus making an accurate measurement difficult. Furthermore, some baseline drift during the trials can occur, therefore the parameter d$t$ was added to Eq 8 as shown in Eq 9. This equation was then used as a model function to estimate through the nonlinear least square's method $\gamma$, $\omega_d$, $A_F$, $B_F$, d$t$ and M. By dividing M by gravity, it was then possible to estimate the mass $m$.

$$F(t) = e^{-\gamma t}(A_F \cos \omega_d t + B_F \sin \omega_d t) + dt + M \tag{9}$$

Subsequently, the "apparent" K and C were obtained from Eqs 10 and 11.

$$K = m\left(\omega_d^2 + \gamma^2\right) \tag{10}$$

$$C = 2m\gamma \tag{11}$$

**2.4.2 - "True" stiffness.** The foot displacements ($\Delta x$ and $\Delta X$) illustrated in (Fig 1c) are related as follows (Eq 12):

$$\Delta X = \frac{r_1}{r_2}\Delta x \tag{12}$$

By considering the moment arms ($r_1$ and $r_2$) illustrated in (Fig 1c), the "true" $k$ and $c$ can be determined. From the condition of dynamic equilibrium of forces in relation to ankle joint, the equation of motion is:

$$\left(I_{foot} + m_{foot}r_{gravity}^2\right)\frac{d^2_\theta}{dt^2} = Fr_1 \cos \theta - fr_2 \cos \theta \tag{13}$$

where $I_{foot}$ is the moment of inertia of the foot in relation to the centre of gravity, $m_{foot}$ the mass of the foot, $r_{gravity}$ is the distance between the centre of rotation and the centre of gravity

of the foot, $\theta$ the instantaneous angular position of the foot, F is the ground reaction force and $f$ the tension of the tendon. Considering that the force that represents the total mass of the system is almost aligned with the axis of rotation, its moment is negligible in absolute terms. Since the summands of the left-hand side of Eq 13 are too small when compared to the ones of the right-hand side, Eq 13 can be approximated to:

$$Fr_1 = fr_2 \tag{14}$$

In this way it is possible to estimate $f$ after measuring the ground reaction force F and determining the distances $r_1$ and $r_2$.

The same rationale used to determine the "apparent" K and C parameters can be used to determine the "true" k and c parameters. Considering the condition of dynamic equilibrium, we obtain Eq 15. Replacing Eqs 12 and 14 in Eq 2 we obtain Eq 16.

$$f = -kx - c\dot{x} \tag{15}$$

$$f = -\left(\frac{r_1}{r_2}\right)^2 Kx - \left(\frac{r_1}{r_2}\right)^2 C\dot{x} \tag{16}$$

Considering that Eqs 15 and 16 are equivalent, the terms of both equations must match, therefore, Eq 17 can be written:

$$k = \left(\frac{r_1}{r_2}\right)^2 K \text{ and } c = \left(\frac{r_1}{r_2}\right)^2 C \tag{17}$$

Hill´s model explain muscle behaviour in terms of the contractile component (CC), the series elastic component (SEC) and the parallel elastic component (PEC), the last one usually disregarded due to its reduced contribution to the overall muscle behaviour [20]. The Hill model depicts a damper and a spring (Fig 2a, drawn in black). However, this spring includes another spring that represents the Achilles tendon elastic behaviour as well as a set of several other parallel springs in series with the Achilles tendon that represents the muscle behaviour of the soleus (Fig 2a, drawn in red). Some of the SEC lies within the CC, particularly in crossbridges, muscle fibres and titin proteins, however a major portion of series elasticity reside in the tendon structure [20, 27]. Since passive tendon stiffness is independent of the CC the stiffness of the SEC can be separated into the parameters $k_d$ and $k_i$ [2, 20]. Most of the stiffness lies in the tendon ($k_t$), which is assumed to be constant [2, 20]. As illustrated in (Fig 2b), $k$ is an increasing function of $f$ and, therefore, it is assumed that the active muscle stiffness of the SEC is proportional to the muscle tension ($k_m = k_d f$). The stiffness $k$ is given by $k_d$ and $k_i$ as in Eq 18, which can be used as a model function to estimate by the nonlinear least squares method $k_m$ and $k_t$ (Fig 2b).

$$k = \frac{k_i k_d f}{k_i + k_d f} = \frac{k_t k_m}{k_t + k_m} \tag{18}$$

**2.4.3 - Displacement ($\Delta$X), angle ($\theta$) and angular velocity ($\omega$) of the foot.** After estimate K and C the velocity and position-time curves were obtained through Simulink software (MATLAB R2013a, MathWorks, Inc., Natick, Massachusetts, United States) as illustrated in (Fig 3). Ankle displacement ($\Delta$X) was then obtained by the position-time curve while the angle ($\theta$) was calculated by basic trigonometry procedures using the variables $\Delta$X and r1 as illustrated in (Fig 1c). Angular velocity ($\omega$) was then taken dividing the angle ($\theta$) by time. More details about procedures and equations used can be found elsewhere [2, 5, 20].

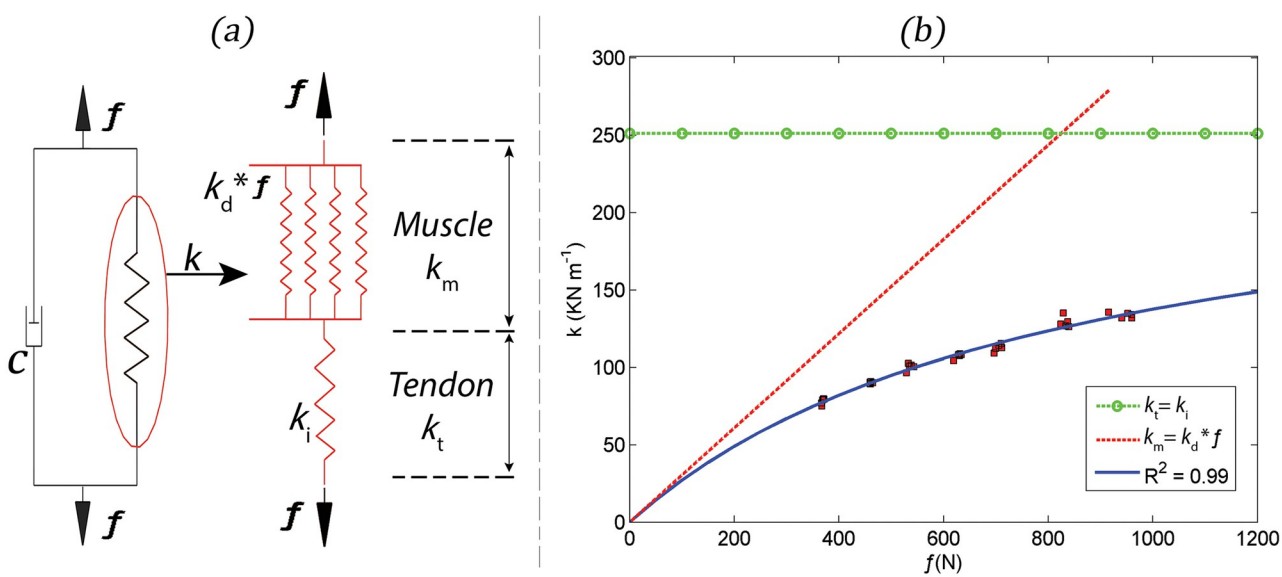

**Fig 2.** Panel (a) shows a schematic representation of the stiffness ($k$) with its constituents ($k_m$ and $k_t$). Considering the ankle's moment arm (Fig 1c), the 'apparent' parameters (i.e., K and C) are transformed into 'true' parameters (i.e., $k$ and $c$). Panel (b) illustrates the k-$f$ curve obtained from the experimental data (i.e., red squares). The ($k_d$) parameter was obtained with the slope of the k-$f$ curve at its origin and used to estimate muscle stiffness ($k_m = k_d f$). Tendon stiffness ($k_t$) was obtained through the horizontal asymptote at high values of the k-$f$ curve.

## 2.5 - Statistical analysis

Statistical analysis was carried out using the Statistical Package for the Social Sciences (IBM SPSS Statistics 28.01, Chicago, IL, USA). A $p$ value $< 0.05$ was considered statistically significant. After evaluating the statistical assumptions (i.e., outliers, normality and sphericity), the analysis of variance (ANOVA) for repeated measures or the Friedman test were conducted to determine the existence of statistically significant differences in the biomechanical properties (i.e., stiffness, damping coefficient, frequency of oscillation, amplitude of movement and angular velocity) for each impulse magnitude applied. When statistically significant differences were found between groups, pairwise comparisons were performed with Bonferroni corrections to control for type1 error rate. Repeated measures ANOVA is robust to deviations from normality, however, if this assumption was violated both the Friedman test and the repeated

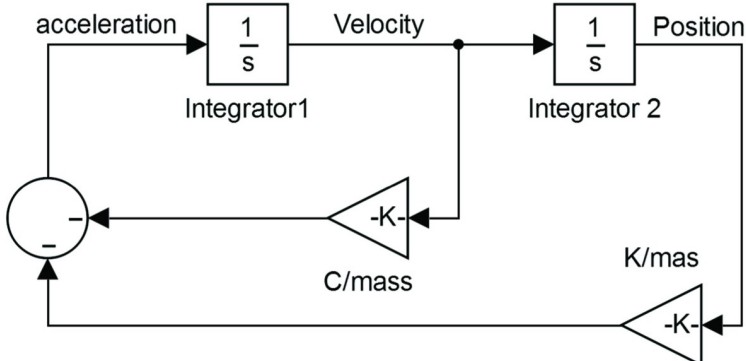

**Fig 3. Model implemented in Simulink software to estimate velocity and position-time curves.**

measures ANOVA were performed. When using Friedman's tests is more appropriate to report the median (Mdn), whereas mean and standard deviation are reported for repeated measure ANOVA. Whenever possible (i.e., when the Friedman's tests and the repeated measures ANOVA gave the same results) only the repeated measures ANOVA were reported. When sphericity assumption was violated, as assessed by the Mauchly's test, then the Huynh-Feldt or Greenhouse-Geisser correction was employed when epsilon ($\varepsilon$) was > 0.75 and < 0.75, respectively [28, 29].

## 3. Results

When considering the "apparent" parameters, the statistical analysis (repeated measures ANOVA) revealed that as the magnitude of the impulse applied increases the stiffness (K) and the natural frequency of oscillation (f) decreased significantly. Post-hoc comparisons further showed that this decrease was significant (p < 0.001) between all the impulses applied. On the other hand, the damping coefficient (C), the amplitude of movement (AM) and the angular velocity ($\omega$) increased significantly (p < 0.001) with the increase of the impulse magnitude (Table 1).

**Table 1. Comparison of stiffness, natural frequency and damping coefficient assessed using three impulses of different magnitude.**

| Variables | | Impulse 1 | Impulse 2 | Impulse 3 | Repeated Measures ANOVA and Friedman Test[F] | Post-hoc with Bonferroni Correction |
|---|---|---|---|---|---|---|
| | | Mean ± SD and Median (Mdn) | Mean ± SD and Median (Mdn) | Mean ± SD and Median (Mdn) | | |
| Total (N = 189) (Loads 10–40 kg) | K (N.m$^{-1}$) | 29224 ± 5087 | 27839 ± 4914 | 26835 ± 4880 | $\chi$2(2) = 17.879, p < 0.001[M]; F(1.850, 347.804) = 87.756, p < 0.001, $\eta$2 = 0.318[H] | 1–2, p < 0.001 |
| | | | | | | 1–3, p < 0.001 |
| | | | | | | 2–3, p < 0.001 |
| | C (N.s/m) | 203 ± 51.0 | 214 ± 56.2 | 222 ± 60.0 | $\chi$2(2) = 25.109, p < 0.001[M]; F(1.793, 336.991) = 22.872, p < 0.001, $\eta$2 = 0.108[H] | 1–2, p = 0.001 |
| | | | | | | 1–3, p < 0.001 |
| | | | | | | 2–3, p < 0.001 |
| | f (Hz) | 4.9 ± 0.5 | 4.8 ± 0.47 | 4.7 ± 0.45 | $\chi$2(2) = 10.154, p < 0.01[M]; F(1.918, 360.670) = 130.777, p < 0.001, $\eta$2 = 0.410[H] | 1–2, p < 0.001 |
| | | | | | | 1–3, p < 0.001 |
| | | | | | | 2–3, p < 0.001 |
| | AM (deg) | 0.66 ± 0.14 | 1.06 ± 0.23 | 1.47 ± 0.32 | $\chi$2(2) = 91.990, p < 0.001[M]; F(1.440, 270.786) = 2196.504, p < 0.001, $\eta$2 = 0.921[G] | 1–2, p < 0.001 |
| | | | | | | 1–3, p < 0.001 |
| | | | | | | 2–3, p < 0.001 |
| | $\omega$ (deg/s) | 6.55 ± 1.73 | 10.25 ± 2.90 | 13.99 ± 3.84 | $\chi$2(2) = 162.57, p < 0.001[M]; F(1.265, 237.857) = 1827.584, p < 0.001, $\eta$2 = 0.907[G] | 1–2, p < 0.001 |
| | | | | | | 1–3, p < 0.001 |
| | | | | | | 2–3, p < 0.001 |
| "True" k (N = 27) (Loads 10–40 kg) | $k_m$ (kN/m)/kN | 630.0 ± 247 | 518.0 ± 208 | 566.0 ± 384 | $\chi$2(2) = 18.667, p < 0.001[F]; | 1–2, p = 0.003 |
| | | Mdn = 564.3 | Mdn = 468.9 | Mdn = 422.2 | | 1–3, p < 0.0005 |
| | $k_t$ (kN/m) | 219.4 ± 81 | 236.5 ± 105 | 234.6 ± 113 | $\chi$2(2) = 1.556, p = 0.459[F]; | ------------------- |
| | | Mdn = 197.4 | Mdn = 210.3 | Mdn = 201.6 | | |

K = "apparent" stiffness, C = "apparent" damping coefficient, f = natural frequency of oscillation, AM = amplitude of movement, $\omega$ = angular velocity, $k_m$ = "true" muscle stiffness, $k_t$ = "true" tendon stiffness, Mdn = Median Values,

[M] the Mauchly's Test of Sphericity,

[H] Repeated Measures Anova with Huynd-Feldt correction,

[G] Repeated Measures Anova with Greenhouse-Geisser correction,

[F] Friedman test.

The statistical analysis (non-parametric Friedman test) of the "true" parameters revealed significant differences only between impulse 1 and 2 and 1 and 3 for $k_m$ but not for $k_t$, as illustrated in Table 1.

## 4. Discussion and conclusion

The aim of the present study was to compare muscle and tendon stiffness outcomes (i.e., "true" stiffness) between the three impulses applied and determine how muscle and tendon are affected by impulse magnitudes. It was hypothesized that different impulse magnitudes would affect the stiffness of the muscle and tendon differently. As expected, the "apparent" stiffness of the whole ankle joint significantly decreased between impulses 1, 2 and 3 (29224 ± 5087 N. m$^{-1}$; 27839 ± 4914 N.m$^{-1}$; 26835 ± 4880 N.m$^{-1}$), respectively. When examining the "true" stiffness, significant differences were found between impulses 1 and 2 (Mdn = 564.3 (kN/m)/kN vs Mdn = 468.9 (kN/m)/kN) and 1 and 3 (Mdn = 564.3 (kN/m)/kN vs Mdn = 422.2 (kN/m)/kN) for muscle (i.e., soleus) stiffness, but not for tendon (i.e., Achilles tendon) stiffness as showed in (Table 1). These results are in line with other studies that used the free-oscillation technique [2, 5, 20] and suggest that the "apparent" stiffness around the ankle joint significantly decreased between impulses due the decrease of the muscle stiffness but not tendon stiffness, which supports the hypothesis initially made. The total stiffness of a muscle results from the sum of the intrinsic stiffness, the reflex mediated stiffness and the passive stiffness [30]. From an isometric state, an external perturbation can lead a muscle to quickly change its length. Short-range stiffness is a biomechanical property that represents the initial muscle response that reflets the intrinsic muscle ability to resist to an external perturbation before any reflex activity or voluntary action take place. Short-range stiffness is mostly due to the resistance associated with the involuntary reversal of the existing cross-bridges in muscle fibres that occur just after the onset of the perturbation [30, 31]. Just after the perturbation, during a short time interval, the muscle has a relatively high stiffness [32–35] that tends to diminish [35–37] when muscle continues to lengthen, as seen with the increases in the impulse magnitudes of the present study. This transition from relatively high to low stiffness is called elastic limit [38] that marks the shift from an elastic muscle behaviour to a more viscous action [34, 35]. The decrease in stiffness is believed to result from the breakage of cross-bridges. The elastic limit for the soleus muscle occurs around an ankle angle of about 1.3 deg [5]. Impulse 2 (range 0.9 to 1.4 deg) approximated this limit while impulse 3 (range 1.2 to 2 deg) surpassed this limit, suggesting cross-bridges breakage could have led to the decreased of $k_m$. Not only does an isolated muscle tend to show greater stiffness when subjected to small rather than large stretches [32, 34]; the muscle sensory receptors (spindle afferents), which are relevant when evoking stretch reflexes, seem to exhibit similar nonlinear amplitude-dependencies, being more sensitive to small amplitude movements [32, 39]. Therefore, the decrease in $k_m$ due the increase in impulse magnitude (and movement amplitude) of the present study probably results from the both the intrinsic and reflex mediated stiffness. The increase of movement amplitude was also accompanied by an increase of velocity (Table 1). It has been reported that the gain of the reflex contribution to stiffness decreases as the average velocity of perturbation increases, which can also help explain the results [40]. Additionally, it has been reported that an increased frequency of motion with added external stiffness, derived from steel springs in parallel with the plantarflexion musculature, can cause an invariant phase delay in reflex response [41]. These phase dependence between motion and reflex response has been reported to influence joint stiffness [42], leading to suggestions that the invariant delay associated with stretch reflex create a phase-interference with system behaviour to reduce stiffness [41]. Rack, Ross [43] also observed this phase-interference behaviour by driving the ankle joint at specific

frequencies. These authors reported that the reflex response was almost in-phase with motion at low frequencies but shifted out-of-phase as the frequency of oscillation increased. The explanation was that with increasing frequencies the reflex delay remains constant but the period of oscillation declines leading the two signals to be in-phase and out-of-phase at different frequencies. In the present study the natural frequency was greater in impulse 1 followed by impulse 2 and 3. Therefore, if the applied impulses of different magnitudes produced a phase-interference with system behaviour the outcome can consist in lower stiffness values due higher impulse magnitudes.

The "true" tendon stiffness ($k_t$) displayed no significant changes across the three impulse magnitudes examined (Table 1). The $k_t$ is usually modelled as a spring in series with the contractile component. However, its stiffness, like most biological materials, is non-linear, with important linear deviations occurring at low tension ranges, i.e., the tension does not rise in direct proportion of the tendon extension [44]. After the initial non-linearities, as the tension increases, stiffness increases until a plateau of constant stiffness is reached [44]. Thus, a spring with constant stiffness is an adequate approximation to model a tendon [2, 20] provided it is stretched beyond the initial region of low stiffness, which was an assumption in the present study as described in section 2.4.2 and illustrated in (Fig 2). It seems therefore that the impulses of the three magnitudes applied in the present study do not affect the estimation of tendon stiffness at this plateau. It has been reported that the error in estimating muscle stiffness equates the moment arm estimation error. A higher degree of sensitivity to changes in moment arms was attributed to measurements of tendon stiffness, particularly at low torque levels [45]; we can't rule out some impact of this in the tendon results.

We have shown that the magnitude of impulse applied affects $k_m$ but not $k_t$. From a practical point of view, practitioners interested in evaluating $k_m$ or the overall apparent stiffness should carefully control the magnitude of impulse applied, especially for comparisons across studies or for multiple measurements over time. On the other hand, if $k_t$ is the focus, the impulse can be applied manually without the need for controlling its magnitude, which will simplify the procedures. This information can assist in refining methods used in the study of either component.

The ability to correctly measure either component individually is relevant both in training and recovery from injury. It is well established that magnitude and timing of tendon and muscle response to mechanical stimuli induced by training are different [46–48]. Similarly, following an injury, the contractile and elastic components of a muscle-tendon complex may have different recovery timelines [49]. There is therefore a practical need to assess them separately for applications in sport training and for monitoring injury recovery.

The contribution of reflex activity and co-activation to stiffness were not assessed in the present study, which is a recognized limitation that should be addressed in future investigations. Another limitation is related to the assumption that the main contributor to musculo-articular stiffness is the soleus and the Achilles tendon, even though other minor contributions may come from passive structures and other muscles [2, 12, 41, 50].

In conclusion, the magnitude of the impulse applied affects both the "apparent" and "true" stiffness. Interestingly, when examining the two components of "true" stiffness, the behavior of $k_m$ follows a similar pattern as "apparent stiffness," whereas $k_t$ seems unaffected by the impulse magnitude. Factors such as the f, AM, ω, and the reflex activity can explain the dependence of $k_m$ on the magnitude of impulse. The assumption made to model $k_t$, and the high sensitivity of tendons to changes in moment arms, can be advocated to explain its behaviour.

## Author Contributions

**Conceptualization:** Aurélio Faria.

**Funding acquisition:** Aurélio Faria.

**Investigation:** Aurélio Faria, Ronaldo Gabriel, Rui Brás, Márcio Soares, Massimiliano Ditroilo.

**Methodology:** Aurélio Faria, Ronaldo Gabriel, Massimiliano Ditroilo.

**Project administration:** Aurélio Faria, Rui Brás.

**Resources:** Aurélio Faria, Rui Brás, Helena Moreira.

**Software:** Aurélio Faria.

**Supervision:** Aurélio Faria, Ronaldo Gabriel, Massimiliano Ditroilo.

**Validation:** Aurélio Faria, Ronaldo Gabriel, Helena Moreira, Massimiliano Ditroilo.

**Visualization:** Helena Moreira, Márcio Soares.

**Writing – original draft:** Aurélio Faria, Márcio Soares.

**Writing – review & editing:** Aurélio Faria, Ronaldo Gabriel, Rui Brás, Helena Moreira, Márcio Soares, Massimiliano Ditroilo.

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
