## [Decision Letter · Decision Letter 0]

14 Mar 2023

PONE-D-23-03008Free-oscillation technique: the effect of the magnitude of the impulse applied on muscle and tendon stiffness around the anklePLOS ONE

Dear Dr. Ditroilo,

Thank you for submitting your manuscript to PLOS ONE. After careful consideration, we feel that it has merit but does not fully meet PLOS ONE’s publication criteria as it currently stands. Therefore, we invite you to submit a revised version of the manuscript that addresses the points raised during the review process.

We look forward to receiving your revised manuscript.

Kind regards,

Yih-Kuen Jan, PhD

Academic Editor

PLOS ONE

Journal Requirements:

Reviewers' comments:

Reviewer's Responses to Questions

**Comments to the Author**

1. Is the manuscript technically sound, and do the data support the conclusions?

Reviewer #1: Yes

Reviewer #2: Partly

2. Has the statistical analysis been performed appropriately and rigorously? 

Reviewer #1: Yes

Reviewer #2: Yes

3. Have the authors made all data underlying the findings in their manuscript fully available?

Reviewer #1: Yes

Reviewer #2: No

4. Is the manuscript presented in an intelligible fashion and written in standard English?

Reviewer #1: Yes

Reviewer #2: Yes

5. Review Comments to the Author

Reviewer #1: General comments

The present study aimed to investigate whether impulses of different magnitudes, applied using the free oscillation technique, could independently influence muscle and tendon stiffness.

In the introduction section, the authors mention the state of the art of the study. But in the discussion and conclusion, they didn’t mention the significant implication for their field, especially for health, rehabilitation, etc.

Since the authors cite the previous study, part of the research method is just short mentioned in this study. Therefore, there is inadequate information on what kind of muscle and tendon was analyzed in this study.

In the discussion section, might the authors add implications in the fields? What is your suggestion for present and future studies?

Specific comments

On the title page, the affiliation seems different.

Department of Sport Science

Department of Sport Sciences

## reviewer: Please author to check it.

Abstract section

The results suggest that the magnitude of the impulse applied influences the whole “apparent” stiffness around the ankle joint. Interestingly, this is driven by muscle stiffness, whereas tendon stiffness appears to be unaffected.

## reviewer: You may suggest providing implications to your fields, for example, health, sport, rehabilitation, etc.

Method section

Stiffness assessment

Briefly, the participant was seated. They had to keep their ankle at 90 deg while holding a load ranging from 10 to 40 kg at 5 kg intervals. A calibrated impulse with a peak force of either 100, 150, or 200 N was randomly applied, and the ensuing damped oscillation was recorded using a force plate/

## reviewer: How long (min) did each subject hold the force?

## reviewer: We found inconsistency in the abbreviation. In the introduction section, the authors mention muscle for km and tendon for kt with lowercase and italics. Still, in another section, the author uses superscripts and subscripts to deal with this.

## reviewer: Please add the references to support your equation.

Results section

It was determined that km and kt differed for each of the three applied impulse magnitudes. However, the non-parametric Friedman test only revealed significant differences between impulses 1 and 2 and 1 and 3 for km but not for kt, as illustrated in Table 1.

## reviewer:

Since your data was ratio, why did the authors choose non-parametric statistical analysis than parametric?

Figure 2

## reviewer: Why did the picture not describe the 90 deg position? It seems more than 90 deg.

Reviewer #2: Summary:

Impulse loads were applied to the foot of healthy individuals to assess the stiffness of the muscles and tendons surrounding the ankle joint. From these trails, mathematical equations were used to determine the stiffness of the muscles and respective tendons surrounding the ankle joint.

Comments:

Abstract: the references to certain abbreviations (Mdn, and KN/m/KN) are not provided, nor are the terms “apparent” stiffness and “true” stiffness well defined

Introduction:

- A reference to the definition of tissue stiffness is needed in the first paragraph

- The purpose is provided, but there is a lack of a hypothesis to drive the research agenda of this project

Methods:

- The authors reference the Faria et al study for the assessment of impulse, but they need to provide additional information to allow the readers in determining how the forces were recorded and measured

- The stiffness assessment is poorly explained. It is suggested that the authors use a diagram to illustrate the procedure for testing the stiffness

- How many trials of the impulse loading was applied, and were these loads randomized to reduce a potential order effect?

- It is also not clear if sufficient rest periods were provided between trails/loads – this is important as repeated loading will modify the stiffness of the tissues

- Since stiffness is the measure that is of most concern, the authors mention ankle joint stiffness and then stiffness of the muscles and tendons around the ankle joint. Which is the stiffness you are truly assessing? Furthermore, were specific muscles assessed, such as the plantar flexors, or was this analysis inclusive of plantar flexor and dorsi flexor muscles?

Results:

- Paragraph 1 does not communicate to the reader well what was reported from the methods. This may be due to the lack of definitions to the abbreviations (although these abbreviations are given in the footer of the table that follows – Table 1)

- Further, the lack of experimental explanation that is expected in the Methods section makes it difficult to clearly interpret the results

Discussion:

- The interpretation of the data, based upon the presented results, should be presented cautiously as many details are not communicated well in the previous sections

- While the theoretical argument brought forward by the authors regarding the stiffness of the muscle versus the tendon is plausible, the data are not as clear – again this is due to a lack of clarity in the Methods section

- What were the limitations of this study? The authors have stated the underlying concepts, but what limits were present that may have influenced the results?

6. PLOS authors have the option to publish the peer review history of their article (what does this mean?). If published, this will include your full peer review and any attached files.

Reviewer #1: **Yes: **Chi-Wen Lung

Reviewer #2: No

---

## [Author Response · Author response to Decision Letter 0]

4 May 2023

REVIEWER 1 

Q1. In the introduction section, the authors mention the state of the art of the study. But in the discussion and conclusion, they didn’t mention the significant implication for their field, especially for health, rehabilitation, etc.

R1. We appreciate the comment of the reviewer. We have added text at the end of the Discussion to address this comment (lines 320-332)

Q2. Since the authors cite the previous study, part of the research method is just short mentioned in this study. Therefore, there is inadequate information on what kind of muscle and tendon was analyzed in this study.

R2. Thank you for the feedback. As suggested by the reviewer we have added more detail (particularly in the Stiffness assessment section) to improve clarity (lines 87-118). Furthermore, in the abstract, introduction, methods and discussion it was also clarified that the Achilles tendon and the soleus were the structures under consideration in the present study (lines 5-8; 50-55, 190-193, 260-263). 

Q3. In the discussion section, might the authors add implications in the fields? What is your suggestion for present and future studies?

R3. As clarified in R1, we have added text at the end of the Discussion to address this comment (lines 320-332)

Specific comments

Q4. On the title page, the affiliation seems different.

 Department of Sport Science

 Department of Sport Sciences

 Please author to check it.

R4. Thank you for pointing this out, the title page was amended. 

Abstract section

The results suggest that the magnitude of the impulse applied influences the whole “apparent” stiffness around the ankle joint. Interestingly, this is driven by muscle stiffness, whereas tendon stiffness appears to be unaffected.

Q5. You may suggest providing implications to your fields, for example, health, sport, rehabilitation, etc.

R5. As clarified in R1, we have added text at the end of the Discussion to address this comment (lines 320-332)

Method section

Stiffness assessment

Briefly, the participant was seated. They had to keep their ankle at 90 deg while holding a load ranging from 10 to 40 kg at 5 kg intervals. A calibrated impulse with a peak force of either 100, 150, or 200 N was randomly applied, and the ensuing damped oscillation was recorded using a force plate/

Q6. How long (min) did each subject hold the force?

R6. Each load was sustained for 10 s, this information was added to the manuscript (line 106).

Q7. We found inconsistency in the abbreviation. In the introduction section, the authors mention muscle for km and tendon for kt with lowercase and italics. Still, in another section, the author uses superscripts and subscripts to deal with this.

R7. Thank you for the comment, each abbreviation was double-checked, they should now all be in order. 

Q8. Please add references to support your equation.

R8. As suggested, additional references were added to support procedures and equations (lines 128 and 210).

Results section

It was determined that km and kt differed for each of the three applied impulse magnitudes. However, the non-parametric Friedman test only revealed significant differences between impulses 1 and 2 and 1 and 3 for km but not for kt, as illustrated in Table 1.

Q9. Since your data was ratio, why did the authors choose non-parametric statistical analysis than parametric?

R9. Thank you for the comment. As mentioned, the data came from a ratio. However, it is a continuous variable, which requires the assumption of normality. As reported in the statistical section, when Friedman's tests (non-parametric test) and the repeated measures ANOVA gave the same results, only the repeated measures ANOVA (parametric test) were reported. Otherwise, the non-parametric Friedman's test was used due to normality issues. 

Figure 2

Q10. Why did the picture not describe the 90 deg position? It seems more than 90 deg.

R10. Fair point, as suggested by the reviewer we have added more detail to improve clarity. In this context, we also adjusted the figures. The measuring position (Figure 1a) shows that the foot-leg, leg-thigh and thigh-trunk angles were set at 90 deg. 

REVIEWER 2 

Reviewer #2: Summary:

Impulse loads were applied to the foot of healthy individuals to assess the stiffness of the muscles and tendons surrounding the ankle joint. From these trails, mathematical equations were used to determine the stiffness of the muscles and respective tendons surrounding the ankle joint.

Comments:

Q1. Abstract: the references to certain abbreviations (Mdn, and KN/m/KN) are not provided, nor are the terms “apparent” stiffness and “true” stiffness well defined

R1. Thank you for the comment. The median (Mdn) abbreviation is now spelled out in the abstract. Regarding the measurement units, kN (kilonewton) and m (metre) are included in the International System of Units (SI), they are well established and widely used, therefore we feel they don’t need to be spelled out, as also reported by Mack 2012 [1]. We have added the definitions of “apparent” and “true” stiffness in the abstract as suggested by the reviewer. The same definitions have been further detailed in the introduction, including appropriate references (lines 4-9, 16, 50-55).

Introduction:

Q2. A reference to the definition of tissue stiffness is needed in the first paragraph.

R2. The concept of stiffness has been defined at the beginning of the introduction (lines 30-32).

Q3. The purpose is provided, but there is a lack of a hypothesis to drive the research agenda of this project.

R3. Thank you for the feedback. As suggested, a hypothesis was introduced alongside the objective (lines 70-73 and 255-258).

Methods:

Q4. The authors reference the Faria et al study for the assessment of impulse, but they need to provide additional information to allow the readers to determine how the forces were recorded and measured.

R4. Thank you for the comment. As suggested by the reviewer we have added more detail (particularly in the Stiffness assessment section) to improve clarity (lines 87-118). 

Q5. The stiffness assessment is poorly explained. It is suggested that the authors use a diagram to illustrate the procedure for testing the stiffness. 

R5. As clarified in previous response, additional details and a new figure (figure 1a) have been added to the methods section to improve clarity (lines 87-118). 

Q6. How many trials of the impulse loading was applied, and were these loads randomized to reduce a potential order effect?

R6. We have added the information that five trials were collected, and that random order was used for both loads and impulses (lines 87-118).

Q7. It is also not clear if sufficient rest periods were provided between trails/loads – this is important as repeated loading will modify the stiffness of the tissues.

R7. This has been clarified in the revised Methods section (lines 87-118). 

Q8. Since stiffness is the measure that is of most concern, the authors mention ankle joint stiffness and then stiffness of the muscles and tendons around the ankle joint. Which is the stiffness you are truly assessing? Furthermore, were specific muscles assessed, such as the plantar flexors, or was this analysis inclusive of plantar flexor and dorsi flexor muscles?

R8. Thank you for the comment. Musculo-articular stiffness (the stiffness around a joint) is usually obtained through the free oscillation technique. This stiffness encompasses muscle, tendons, ligaments, joint capsule, etc. However, muscle and tendon are the predominant factors [2-4]. We assessed the “apparent” stiffness (musculo-articular stiffness) around the ankle joint. Additionally, we also assessed the “true” stiffness, which is the stiffness of the muscle (particularly the soleus) and tendon (Achilles tendon). In the “apparent” approach, we have the whole stiffness (mainly from the tendon and muscle), while in the true stiffness, we separate the muscle from the tendon. Furthermore, in the assessment position (knee at 90 deg) the main contributor to stiffness is the soleus (as reported in the literature). Other muscles may play a minor role, but their contribution is usually negligible. This information was clarified in the abstract, introduction and methods, and discussion sections (lines 2-10; 35-37; 50-58; 114-118; 190-193; 333-337). 

Results:

Q9. Paragraph 1 does not communicate to the reader well what was reported from the methods. This may be due to the lack of definitions to the abbreviations (although these abbreviations are given in the footer of the table that follows – Table 1). Further, the lack of experimental explanation that is expected in the Methods section makes it difficult to clearly interpret the results

R9. Thank you for the comment. As clarified previously, the methods section has been expanded to include additional details. The results section has also been amended. As a result, we believe clarity has improved significantly. 

Discussion:

Q10. The interpretation of the data, based upon the presented results, should be presented cautiously as many details are not communicated well in the previous sections. While the theoretical argument brought forward by the authors regarding the stiffness of the muscle versus the tendon is plausible, the data are not as clear – again this is due to a lack of clarity in the Methods section.

R9. We appreciated the comment of the reviewer. As suggested, improvements have been made to several sections, particularly to the methods and results section. Some other adjustments were also made in the discussion. As a result, we believe the manuscript is improved for clarity. 

Q11. What were the limitations of this study? The authors have stated the underlying concepts, but what limits were present that may have influenced the results?

R11. Thank you for pointing this out. Limitations have now been added to the manuscript at the end of the Discussion (lines 333-337).

References:

1. Mack C. How to write a good scientific paper: Acronyms Journal of Micro/Nanolithography MEMS, and MOEMS. 2012.

2. Faria A, Gabriel R, Abrantes J, Brás R, Moreira H. Triceps-surae musculotendinous stiffness: relative differences between obese and non-obese postmenopausal women. Clin Biomech. 2009;24(10):866–71. Epub 2009/08/26. doi: 10.1016/j.clinbiomech.2009.07.015. PubMed PMID: 19703726.

3. Ditroilo M, Watsford M, Murphy A, De Vito G. Assessing musculo-articular stiffness using free oscillations theory, measurement and analysis. Sports Med. 2011;41(12):1019-32. PubMed PMID: ISI:000297614200003.

4. Butler RJ, Crowell III HP, Davis IM. Lower extremity stiffness: implications for performance and injury. Clin Biomech. 2003;18(6):511-7. Epub 2003/06/28. PubMed PMID: 12828900.

---

## [Decision Letter · Decision Letter 1]

24 May 2023

Free-oscillation technique: the effect of the magnitude of the impulse applied on muscle and tendon stiffness around the ankle

PONE-D-23-03008R1

Dear Dr. Ditroilo,

We’re pleased to inform you that your manuscript has been judged scientifically suitable for publication and will be formally accepted for publication once it meets all outstanding technical requirements.

Kind regards,

Yih-Kuen Jan, PhD

Academic Editor

PLOS ONE

Additional Editor Comments (optional):

Reviewers' comments:

Reviewer's Responses to Questions

**Comments to the Author**

1. If the authors have adequately addressed your comments raised in a previous round of review and you feel that this manuscript is now acceptable for publication, you may indicate that here to bypass the “Comments to the Author” section, enter your conflict of interest statement in the “Confidential to Editor” section, and submit your "Accept" recommendation.

Reviewer #2: All comments have been addressed

2. Is the manuscript technically sound, and do the data support the conclusions?

Reviewer #2: Yes

3. Has the statistical analysis been performed appropriately and rigorously? 

Reviewer #2: Yes

4. Have the authors made all data underlying the findings in their manuscript fully available?

Reviewer #2: Yes

5. Is the manuscript presented in an intelligible fashion and written in standard English?

Reviewer #2: Yes

6. Review Comments to the Author

Reviewer #2: (No Response)

7. PLOS authors have the option to publish the peer review history of their article (what does this mean?). If published, this will include your full peer review and any attached files.

Reviewer #2: No

---

## [Editor Report · Acceptance letter]

7 Jun 2023

PONE-D-23-03008R1 

Free-oscillation technique: the effect of the magnitude of the impulse applied on muscle and tendon stiffness around the ankle 

Dear Dr. Ditroilo:

I'm pleased to inform you that your manuscript has been deemed suitable for publication in PLOS ONE. Congratulations! Your manuscript is now with our production department. 

Kind regards, 

on behalf of

Dr. Yih-Kuen Jan 

Academic Editor

PLOS ONE